# Apple Root Microbiome as Indicator of Plant Adaptation to Apple Replant Diseased Soils

**DOI:** 10.3390/microorganisms11061372

**Published:** 2023-05-24

**Authors:** Nivethika Ajeethan, Shawkat Ali, Keith D. Fuller, Lord Abbey, Svetlana N. Yurgel

**Affiliations:** 1Department of Plant, Food, and Environmental Sciences, Dalhousie University, Truro NS B2N 5E3, Canada; 2Department of Biosystems Technology, Faculty of Technology, University of Jaffna, Kilinochchi 44000, Sri Lanka; 3Agriculture and Agri-Food Canada, Kentville Research and Development Centre, Kentville NS B4N 1J5, Canada; 4USDA, ARS, Grain Legume Genetics and Physiology Research Unit, Prosser, WA 99350, USA

**Keywords:** specific replant disease, apple replant disease, apple microbiome, plant growth promoting microbes

## Abstract

The tree fruit industry in Nova Scotia, Canada, is dominated by the apple (*Malus domestica*) sector. However, the sector is faced with numerous challenges, including apple replant disease (ARD), which is a well-known problem in areas with intensive apple cultivation. A study was performed using 16S rRNA/18S rRNA and 16S rRNA/ITS2 amplicon sequencing to assess soil- and root-associated microbiomes, respectively, from mature apple orchards and soil microbiomes alone from uncultivated soil. The results indicated significant (*p* < 0.05) differences in soil microbial community structure and composition between uncultivated soil and cultivated apple orchard soil. We identified an increase in the number of potential pathogens in the orchard soil compared to uncultivated soil. At the same time, we detected a significant (*p* < 0.05) increase in relative abundances of several potential plant-growth-promoting or biocontrol microorganisms and non-fungal eukaryotes capable of promoting the proliferation of bacterial biocontrol agents in orchard soils. Additionally, the apple roots accumulated several potential PGP bacteria from Proteobacteria and Actinobacteria phyla, while the relative abundances of fungal taxa with the potential to contribute to ARD, such as Nectriaceae and plant pathogenic *Fusarium* spp., were decreased in the apple root microbiome compared to the soil microbiome. The results suggest that the health of a mature apple tree can be ascribed to a complex interaction between potential pathogenic and plant growth-promoting microorganisms in the soil and on apple roots.

## 1. Introduction

Replant disease is a global problem in tree fruit cultivation and is common in Nova Scotia (NS), Canada. Specific replant disease (SRD) is a soil-borne disease that causes morpho-physiological reactions in plants following the replanting of the same plant species at the same site. SRD has been reported for a broad spectrum of plants from the family Rosaceae, such as apple (*Malus domestica*), pear (*Pyrus communis*), cherry (*Prunus avium*), peach (*Prunus persica*), strawberry (*Fragaria* × *ananassa*), and rose (*Rosa rubiginosa*) [1]. SRD is characterized by poor tree growth and low productivity of newly planted apple trees in old orchard soils [2]. The development and severity of SRD are dependent on plant vigor, physiological state, and biotic factors (root- and soil-associated microbiomes), which can be aggravated by abiotic stresses such as plant water status, temperature, and soil fertility [3]. SRD symptoms include uneven growth throughout the orchard and stunting and shortening of internodes of the affected trees [4]. SRD-affected plant roots showed discoloration, root tip necrosis, and a general reduction in root biomass [1]. A delay in initial fruit production, shortened internodes, and a general reduction in overall fruit yield and quality are the most evident symptoms.

The etiology of SRD-affected apple trees, also known as apple replant disease (ARD), is complicated to define for several reasons. ARD is associated with a pathogen complex, which is a microbial consortium of soil-borne pathogenic fungi *Cylindrocarpon*, *Rhizoctonia*, and oomycetes such as *Phytophthora* and *Pythium* [5,6,7]. Root lesion nematode (*Pratylenchus penetrans*) capable of causing a persistent damage to a variety of crops can also cause ARD [8]. While considerable agreement in the literature exists on causal agents of ARD, there is also growing awareness that the presence of one or more dominant players may exist. Furthermore, such dominant players may vary amongst replant sites and soil types [4]. Moreover, like other plant diseases, ARD disease occurrence is often associated with changes in microbiome structure, diversity, and function [9,10,11], which can influence microbial domination.

Multiple strategies have been applied to comprehensively understand ARD etiology. Culturing of apple trees can trigger changes in indigenous soil microbiome that may lead to disease development [4,12]. For example, the lack of beneficial groups of bacteria in orchard soil was proposed to aggravate the effects of plant pathogens [13]. Additionally, the formation of root-derived phenolic compounds and root debris in the soil might lead to the selection of specific microbes and the accumulation of microbial pathogens among the orchard soil microbiomes. It was reported that plant metabolites could determine soil microbiome composition and function, which contributes to plant protection from pathogens and nutrient uptake [14,15], and those plant metabolites can differ among apple rootstock genotypes [11]. Many studies suggested that besides fungi and oomycetes, other soil-borne organisms could be key players in ARD development (e.g., bacteria and nematodes), either by facilitating or discouraging pathogen establishment or by being pathogenic or beneficial themselves [16,17].

On the other hand, host plants manipulate their associated microbiomes through the provision of selective environments in their rhizosphere. These selective environments stimulate the accumulation of plant growth-promoting (PGP) bacteria inside root tissues and in root-associated soils [18]. PGP bacteria provide nutrients such as nitrogen, phosphate, and iron, and facilitate plant growth and development. They can also stimulate plant growth or prevent diseases by producing indole-3-acetic acid (IAA), gibberellins, and cytokinins, or/and exhibit 1-aminocyclopropane-1-carboxylic acid (ACC) deaminase activity [19].

Management practices to control ARD can be costly, inefficient, and in many cases, detrimental to human and environmental health, and soil sustainability [20]. Soil microbiomes play a significant role in soil health and plant disease prevention [21], and as a result, promising approaches to increase soil disease suppression include direct and indirect manipulations of soil- and root-associated microbial communities. The present research was triggered by the hypothesis that a more favorable equilibrium between soil pathogenic microbial species accumulated during the life of an apple orchard and the beneficial microbiome in their roots might keep replanted young apple trees healthier in old orchard soils. Therefore, it is important to characterize the soil- and root-associated microbiomes in mature orchards by surveying the accumulation of pathogenic and beneficial taxa that can participate in this equilibrium.

## 2. Materials and Methods

### 2.1. Description and Sampling of Sites

Uncultivated bulk soil samples (pH 4.7) were collected previously in August 2015 from two undisturbed forest sites adjacent to Dalhousie University Research Centre fields in Debert, NS, (45°26′29″ N 63°27′2″ W and 45°26′36″ N 63°27′1″ W) and Collingwood, NS, (45°36′53″ N 63°56′32″ W) [22]. The methods of collection, preparation, processing, and sequencing of these samples were previously described [22]. Cultivated orchard bulk soil and apple tree root samples were collected between September 2019 and September 2020 from six mature apple orchards located in the Annapolis Valley, NS. Sampling location and age of orchard were Orchard_1 (45°5′31″ E 64°28′31″ W) at 100 yrs; Orchard_2 (45°2′4″ E 64°43′4″ W) at 50 yrs; Orchard_3 (45°0′53″ E 64°42′36″ W) at 30 yrs; Orchard_4 (45°4′37″ N 64°23′56″ W) at 30 yrs; Orchard_5 at 45 yrs; Orchard_6 at 44 yrs.

A total of 36 apple tree root and 36 soil samples (6 sub-samples of root and 6 sub-samples of soil per orchard) were collected from a depth of 0 to 30 cm close to the fine roots and another set of samples ~1.5 m away from the mature tree trunks. The soils were sieved through a 5-mm sieve and immediately placed in sterile bags and transported to the laboratory on ice for storage at −20 °C. Additionally, 5-g soil samples were sieved through a 2-mm mesh and stored at −80 °C for DNA extraction. Around 20 g of roots were collected from the same soil sampling locations. The roots were placed in sterile bags and transported to the laboratory on ice. The roots were washed 3 times with 10% glycerol and sonicated 3 times as described by White [23] before storing at −80 °C for further analysis. The frozen roots were ground into fine powder in liquid nitrogen, and 0.25 g of root tissue was set aside for DNA extraction.

### 2.2. DNA Extraction and Sequencing

Soil and plant DNA extraction was carried out using the Omega Bio-Tek DNA extraction kit according to the manufacturer’s protocol (Omega Bio-Tek, Inc., Norcross, Georgia). DNA quality and concentration were measured using Bio-Tek Synergy H1 Hybrid Multi-Mode Reader and the Gen5 software (version 3.08) application. At least 50 ng (10 µL) of DNA samples were sent to the Dalhousie University Centre for Comparative Genomics and Evolutionary Bioinformatics–Integrated Microbiome Resource (CGEB-IMR) for V6–V8 16S rRNA gene (16S; forward: 5′-ACGCGHNRAACCTTACC-3′; reverse: 5′-ACGGGCRGTGWGTRCAA-3′), V4 18S rRNA gene (18S; forward: 5′-CYGCGGTAATTCCAGCTC-3′; reverse: 5′-AYGGTATCTRATCRTCTTYG-3′) and ITS2 region (ITS2; forward: 5′-GTGAATCATCGAATCTTTGAA-3′; reverse: 5′-TCCTCCGCTTATTGATATGC-3′) [21,22] library preparation and sequencing. The 16S/ITS2/18S and 16S/ITS2 amplicon sequencing were used to study soil and apple roots microbiomes, respectively. Samples were multiplexed using a dual-indexing approach and sequenced using an Illumina MiSeq with paired-end 300 + 300 bp reads [24]. All PCR procedures and Illumina sequencing details were previously described [22,25].

### 2.3. 16S rRNA, 18S rRNA, and ITS2 Amplicon Sequence Processing

The 16S and 18S reads of the uncultivated soil obtained from the previous study [22] were combined with the 16S and 18S reads obtained in the present study. The sequence processing was performed using the standard operating procedure as outlined in the Microbiome Helper package [25]. For 16S reads, the sequences were trimmed of their primers using QIIME2’s Cutadept plug-in [25], and then the overlapping paired-end forward and reverse reads were stitched together using the QIIME2 VSEARCH wrapper [26]. The 18S and ITS2 reads were stitched together using PEAR [27], and then the sequences were trimmed of their primers using QIIME2’s Cutadept plug-in [25]. Sequences were filtered for low-quality or probable chimeric reads from the dataset by using QIIME2’s q-score-joined function (QIIME2 version 2020.8). Using QIIME2’s Deblur plug-in, the sequences were organized into amplicon sequence variants (ASVs) high-resolution genomic groupings [25]. Taxonomic classifications were assigned to the ASV using QIIME2’s naive-Bayes approach implemented in the scikit learn function, referencing SILVA databases [28]. Furthermore, low-abundant ASVs were removed, and ASVs assigned to mitochondria and chloroplasts were also filtered out [25].

### 2.4. Bioinformatics and Statistical Analysis

Identification of differentially represented taxa were conducted using STAMP software (version 2.1.3) and the Welch’s *t*-test to identify which taxa relative abundance varied significantly (*p* < 0.05) between niches (apple roots/orchard soil/uncultivated soil) [29] and adjusted *p*-values were calculated using the Benjamini–Hochberg FDR multiple-test correction. Relative abundances of bacterial and fungal taxonomic groups were represented as a percentage (%) of their niches for the respective 16S, 18S, or ITS2 reads if not indicated otherwise. QIIME2’s diversity function was used to calculate alpha diversity (Shannon indices) and beta diversity (UniFrac matrices) [30,31]. Bray–Curtis distance matrices were then subjected to an analysis of variance using distance matrices statistical method (ADONIS test), through which their values were fitted to a linear regression to determine what proportion of variance in community structure could be attributed to different niches. Non-metric multidimensional scaling (NMDS) of bacterial communities was performed on Bray–Curtis matrices using Vegan R package [32,33]. The graphics were created through RStudio (Version 1.2.5001, RStudio Inc., Boston, MA, USA) using the qiime2R package and plotted using ggplot2 [34].

## 3. Results

### 3.1. Data Description

Bacterial community: After filtering unclassified and plant-derived mitochondria and chloroplasts ASVs, the 16S dataset contained a total of 1,744,577 reads and 7361 features across 81 samples (36 apple root, 36 orchard soil, and 17 uncultivated soils) with a mean frequency of 21,538 reads per sample, and a median frequency of 20,009 reads per samples. The 16S samples were rarefied to a depth of 5900 reads per sample for a total of 477,900 reads and 7360 features (Appendix A). These features were distributed across 29 bacterial phyla where Actinobacteriota (33%), Proteobacteria (32%), Bacteroidota (9%), Acidobacteriota (8%), and Myxococcota (5%) were the top five most relatively abundant bacterial phyla in total apple microbiome. Proteobacteria and Actinobacteriota were the most relatively abundant bacterial phyla in apple roots (37% and 36% of apple roots’ 16S reads, respectively) and in orchard soil (31% and 27%, respectively). Acidobacteriota, Proteobacteria, Verrucomicrobiota, Chloroflexi, Bacteroidota, and Actinobacteriota (47%, 20%, 10%, 6%, 5%, and 4%, respectively) were the most relatively abundant bacterial phyla in the uncultivated soil (Figure 1A).

Fungal community: The ITS2 dataset contained a total of 8,003,297 reads and 1050 features spread across 72 samples (36 orchard soil and 36 apple root), with a mean frequency of 111,157 reads per sample, a median frequency of 88,185 reads per sample after filtering unclassified and plant-derived ASVs. For normalization purposes, the ITS2 samples were rarefied to a depth of 7190 reads per sample for a total of 503,300 reads and 1050 features, and after the rarefaction processes had been performed, the two low-depth samples were removed from the ITS2 dataset (Appendix A). ITS2 features were spread across 13 fungal phyla and 40 classes. Ascomycota, Basidiomycota, Mortierellomycota, and Glomeromycota were the most relatively abundant fungal taxa in the apple orchard total microbiome (Appendix A) and were represented by 80%, 13%, 5%, and 1%, respectively. In the orchard soil Ascomycota (67%), Basidiomycota (22%), and Mortierellomycota (10%) were the most abundant fungal phyla (Appendix A), including Ascomycota classes Sordariomycetes, Leotiomycetes, and Dothideomycetes and Basidiomycota Tremellomycetes were represented by 29%, 16%, 14%, and 20%, respectively (Figure 1B). Ascomycota (93%), Basidiomycota (5%), and Glomeromycota (2%) were the most abundant in apple roots (Appendix A), including fungal classes Ascomycota Dothideomycetes, Leotiomycetes, and Sordariomycetes, which were represented by 52%, 27% and 11%, respectively (Figure 1B).

Eukaryotic community: After filtering unclassified and plant-derived ASVs, the 18S dataset had a total of 308,956 reads and 3308 features spread across 53 samples (i.e., 36 orchard soil and 17 uncultivated soil samples) with a mean frequency of 5829 reads per sample, and a median frequency of 5302 reads per sample. The 18S samples were rarefied to a depth of 1434 reads per sample for a total of 76,002 reads and 3149 features (Appendix A). In the orchard soil, Ascomycota (29%), Cercozoa (17%), Chlorophyta (13%), and Phragmoplastophyta (11%) were the most abundant eukaryotic phyla (Figure 1C). Dinoflagellata (12%), Tunicata (12%), Ascomycota (12%), and Phragmoplastophyta (11%) were the most abundant in the uncultivated soil. (Figure 1C).

### 3.2. Effect of Soil Location on Bacterial and Eukaryotic Microbiomes

Bacterial community: NMDS visualization demonstrated a strong separation in bacterial community structure between the orchard and uncultivated soils (Figure 2A). The strength and statistical significance of sample groupings indicated that the niche was a significant factor in shaping the soil microbiome. Around 39% bacterial community variation (*p* < 0.01) was observed between the orchard and uncultivated soil microbiome (Table 1). Furthermore, we detected a significant increase (*p* < 0.05) in bacterial Shannon diversity in the orchard soil (9.3) when compared with the uncultivated soil (7.8) (Figure 2B). Fifty-six bacterial classes were differentially represented between the orchard and uncultivated soil microbiomes (Appendix A). The classes represented by at least 3% of their respective soils’ 16S reads and overrepresented in the orchard soil included Alphaproteobacteria, Thermoleophilia, Actinobacteria, Vicinamibacteria, Actinobacteriota MB-A2-108, Chloroflexi KD4-96, Polyangia, Gemmatimonadetes, and Acidimicrobiia. Acidobacteriae and Verrucomicrobiae were under-represented when compared with the uncultivated soil (Appendix A). The most abundant bacterial genera that were overrepresented in orchard soil included *Pseudolabrys,* Actinobacteriota *Streptomyces*, *Nocardioides*, MB-A2-108 and 67-14, Chloroflexi KD4-96, *Gaiella*, *Mycobacterium*, *Nakamurella*, and *Bacillus* as well as unclassified Gaiellales, Vicinamibacterales, and Gemmatimonadaceae. On the other hand, the relative abundances of Acidobacteriae Subgroup-2 and *Bryobacter*, unclassified Acidobacteriales, and *Xiphinematobacter* were significantly (*p* < 0.05) decreased in orchard soil (Figure 2C).

Eukaryotic community: There was a strong separation in the NMDS plot between the orchard soil microbiome and the uncultivated soil microbiome (Figure 3A). According to Kruskal–Wallis test, there was no significant difference (*p* > 0.05) in Shannon diversity of the eukaryotic community between the orchard and the uncultivated soils (Figure 3B). However, the analysis of variance showed a significant difference between sampling niches. In total, 30% of eukaryotic community variation (*p* < 0.01) was observed between the orchard soil microbiome and the uncultivated soil microbiome (Table 1). This strong community variation was accompanied by a significant difference (*p* < 0.05) in eukaryotic community composition. More specifically, 75 eukaryotic classes were differentially represented. (Appendix A). These included highly abundant Sordariomycetes, Chlorophyceae, Glissomonadida, Leotiomycetes, Dothideomycetes, and Cercomonadidae, with Appendicularia, Dinophyceae, Prymnesiophyceae, Acantharia, and Archaeorhizomycetes being relatively under-represented in orchard soil when compared with uncultivated soil (Appendix A).

Considering eukaryotic genera, *Archaeorhizomyces*, *Phaeocystis,* unclassified Oikopleuridae, Gymnodiniphycidae, Polycladida, Arthracanthida, and Dinoflagellata were only found in the uncultivated soil, while genera *Heteromita*, *Trichoderma*, *Cercomonas*, and unclassified genera from Chlorophyceae, Leotiomycetes, Hypocreales, Cercozoa, Trebouxiophyceae, Dorylaimida, Pleosporales, Sordariales, Plectosphaerellaceae, Herpotrichiellaceae, Helotiales and Klebsormidiophyceae were overrepresented in orchard soil (Figure 3C). Importantly, two of the causal agents of ARD, the oomycetes *Pythium*, *Phytophthora,* and *Fusarium,* were only present in the orchard soils [5,6,7] (Appendix A).

### 3.3. Differences between Microbial Communities of Apple Root and Orchard Soil

Bacterial community: NMDS analysis demonstrated a strong visual separation in bacterial community structure between apple root and orchard soil (Figure 4A). Statistical analysis of sample grouping indicated a significant (*p* < 0.01) difference between sampling niches with 31% bacterial community variation between apple roots microbiome and orchard soil microbiome (Table 1). Additionally, a significant (*p* < 0.05) difference in Shannon diversity was detected between bacterial communities from apple roots and those from the orchard soil (Figure 4B). We also detected a significant variation in bacterial relative abundances between root and soil environments. In total, 74 bacterial classes were differentially represented between root and soil microbiomes (Appendix A). Among the relatively abundant classes, Actinobacteria, Alphaproteobacteria, Gammaproteobacteria, Bacteroidia, Polyangia, and Acidimicrobiia were overrepresented, while Thermoleophilia, Vicinamibacteria, Actinobacteriota MB-A2-108, and Acidobacteriae were underrepresented in apple root (Appendix A). Several genera with a potential to improve host plant growth were over-represented in apple roots, including Actinobacteria *Streptomyces*, Micromonosporaceae, *Actinoplanes*, and *Nonomuraea*, Proteobacteria *Bradyrhizobium*, *Rhizobium* group, *Steroidobacter*, *Sandaracinaceae*, Rhodanobacteraceae, *Phenylobacterium*, *Dongia* and *Sphingomonas, and* Bacteroidia *Niastella* and Chitinophagaceae (Figure 4C).

Fungal community: dissimilarity between niches visualized in NMDS plots showed a strong separation between apple root and bulk soil microbiomes (Figure 5A), which was confirmed by analysis of variance indicating that the niche was a significant factor shaping fungal community (R^2^ = 0.389, *p* < 0.01) (Table 1). We also detected a significant (*p* < 0.05) decrease in fungal Shannon diversity in apple roots (2.7) compared to orchard soil (6.0) (Figure 5B). Additionally, 21 fungal classes were differentially represented between root and soil microbiome (Appendix A), including highly abundant Dothideomycetes and Leotiomycetes and poorly represented Tremellomycetes, Mortierellomycetes, Saccharomycetes, and Eurotiomycetes in the roots, when compared with orchard soil (Appendix A). The relative abundances of unclassified genera from orders Pleosporales and Helotiales and *unclassified Sordariomycetes* were significantly (*p* < 0.05) increased, and the relative abundances of fungal taxa with the potential to contribute to ARD, such as *Fusarium* and Nectriaceae, were decreased in apple root microbiome, compared to soil microbiome (Figure 5C).

### 3.4. Variation of Soil and Apple Root Microbiome across the Orchards

Location was a significant factor affecting orchard soil and root microbiome. It explained 40% of bacterial, 42% of fungal, and 34% of eukaryotic soil community variation (*p* < 0.001), and 36% of bacterial and 31% of fungal apple root community variations were explained between the locations (Table 1). Additionally, the soil and root microbiomes differ in their Shannon diversity between the locations. For example, we found that Orchard_5 had the lowest fungal and eukaryotic Shannon diversity, while Orchard_1 had the lowest bacterial Shannon diversity (Appendix A). We also detected that several microbial taxa were differentially represented in the soil (Appendix A), and the root microbiomes across six different orchards (Appendix A). However, none of the potential members of the ARD pathogenic complex (*Pythium*, *Phytophthora)* differed in their relative abundances between the orchard soils. On the other hand, potential plant growth-promoting genera, *Trichoderma*, was overrepresented in Orchard_3 and Orchard_1 (Figure 6).

## 4. Discussion

Plants are sessile organisms and are constantly exposed to environmental stresses. However, they can mitigate the effect of these stresses by deploying environmental microbiomes for their protection [35]. Host plants continuously shape root-associated microbial communities according to changes in biotic and abiotic factors [3,36,37,38]. Here, we present a profile of microbiota potentially associated with ARD to better understand the cause of the disease in mature orchards. We also analyzed apple root microbiomes to understand the mechanisms of host-plant adaptation to hostile biotic soil environments. 

Orchard soil microbiome: Long-term cultivation practices such as application of herbicides, fertilization, chopping of pruning and mowing of laneway vegetation, and other practices can influence the assemblage of a soil microbiome [39]. Moreover, microbiota associated with different plant genotypes can differ considerably [36]. Previously, uncultivated soil samples were collected, processed, and sequenced [22]. There might be some minor differences that could slightly affect the ASV annotation. However, this factor will not introduce changes in the uncultivated soil microbiome comparable to the differences that resulted from factors such as soil origin or management. When considering the differences between the orchard and uncultivated soil, it is also important to recognize that soil properties altered by orchard management practices such as tillage at the time of establishment, application of lime and fertilizers, use of herbicides and the spraying of agro-chemicals, will in all likelihood alter soil and root microbiomes over time. For example, soil pH might influence the microbiome [40]. This study focused on the biological characteristics of orchard soils (when compared with uncultivated soils), in particular the presence of plant growth-promoting and pathogenic microorganisms, as it relates to ARD.

Similar to previous studies [22,40], our results indicated that the origin of bulk soil (i.e., orchard vs. uncultivated) was a significant factor in shaping soil bacterial and eukaryotic communities. These results confirm that orchard management contributed to the changes in microbiome structure and composition in the soils. However, the persistence of specific plant species, such as apple trees in orchards, in contrast to highly diverse forest plant communities, might also be an important factor in shaping soil microbiomes [41]. 

We identified several bacterial taxa with the potential to improve host plant growth and soil fertility, which were over-represented in the orchard soil. These include *Pseudolabrys and Bacillus*, as well as several Actinobacteria taxa. *Pseudonocardia* members can produce IAA as well as other secondary metabolites that exhibit anti-bacterial, anti-fungal, and anti-viral activities and promote plant growth [42]. *Bacillus* spp. support plant growth by phosphate and potassium solubilization, siderophores production, nitrogen fixation, and phytohormone synthesis. *Bacillus* could act as a biocontrol agent by secreting antibiotics and lytic enzymes and demonstrate antagonistic activity against phytopathogens [43,44]. Actinobacteria play a critical role in recycling organic matter, improving soil carbon and nitrogen availability, and possess a number of plant growth-promoting traits [45]. More specifically, some *Nocardioides* species possess antagonistic activities that provide defense against phytopathogens through the production of antimicrobial compounds. For example, *Nocardioides* have biocontrol activities against wheat (*Triticum aestivum)* pathogen, *Rhizoctonia solani* [46].

Several eukaryotic taxa, such as fungi *Trichoderma* and bacterivorous *Cercomonas* and *Heteromita* with potential biocontrol functions were overpopulated in the orchard soil. A powerful biostimulant and biocontrol agent, *Trichoderma* [47,48,49], was 15-fold more abundant in orchard soils when compared with the uncultivated soils, although their relative abundance varied significantly between orchard soils. The predatory soil flagellate from the *Cercomonas* genus promotes the proliferation of highly toxic bacterial biocontrol agents such as *Pseudomonas* or *Bacillus* by targeting fewer toxic bacteria, while *Heteromita* taxa promote the proliferation of toluene biodegrading *Pseudomonas* spp. [50,51,52]. Additionally, *Mrakia* and *Metarhizium* were highly abundant in orchard soils. *Metarhizium* spp. are known biocontrol agents against insects, arachnids, and other arthropod pests [53], and could help improve phosphate content in soil [54]. *Mrakia* species could promote plant growth by phosphate solubilization at low temperatures and inhibit the growth of the plant pathogen *Alternaria solani* [55]. 

On the other hand, several potential phytopathogens (*Fusarium* spp.) and causal agents of ARD, including *Pythium* and *Phytophthora,* were only found in the orchard soil. Overwintering structures (oospores) produced by *Phytophthora* and *Pythium* can survive in dead or dormant roots and orchard soil. When new trees are planted, overwintering oospores become active, and *Pythium* spp. cause severe damage to young trees by stripping root hairs resulting in a reduction in water and nutrient uptake [56]. Furthermore, *Fusarium* spp. also has been identified in replanted orchard soils [57]. Additionally, our data indicate an increase in the relative abundance of free-living nematode Dorylaimida in orchard soils. It was previously reported that Dorylaimida was more abundant in ARD soils, and its abundance was significantly correlated with root growth reduction [58], suggesting its potential role in the ARD pathogenic complex. Overall, our soil microbiome analysis indicated significant differences in microbial community structure and composition between uncultivated and cultivated orchard environments. 

We identified an increase in the number of potential pathogens, which either belong to known ARD pathogenic complexes or have the potential to facilitate ARD development. At the same time, we detected a significant increase in relative abundances of several potential plant growth-promoting or biocontrol microorganisms and in non-fungal eukaryotes capable of promoting the proliferation of bacterial biocontrol agents in orchard soils. This is an indication that while mature apple trees cannot completely defend themselves against root pathogens, they are able to mitigate their impact through the manipulation of the rhizosphere composition to better defend themselves. This is a possible reason why the worst expression of replant disease in apple trees most often occurs in the first two to three growing seasons after planting [59], while the root system of the young trees on clonal rootstocks builds their own defense mechanisms. Our data suggested a much more complex interaction between soil microbiome and apple tree. 

Apple root microbiome: Our results indicate a significant variation in bacterial and fungal communities’ structure, composition, and diversity between root and soil microbiomes. The difference in microbiome diversity in soil and root might be the result of increased interactions within microbiomes and the microbial species selection process across the soil–root continuum [22,60,61]. Our analysis showed a decrease in both fungal and bacterial Shannon diversity in roots compared to orchard soils. This is in agreement with previous studies, which suggest that a decrease in bacterial alpha diversity in tree roots is the result of a major plant selective pressure [22,62].

The differences in microbial taxonomic composition between roots and soil microbiomes are well documented [62,63,64]. In our study, Proteobacteria, Actinobacteria, and Bacteroidetes were overrepresented, while Acidobacteriota, Chloroflexi, Verrucomicrobiota, and Firmicutes were underrepresented in apple root when compared with bulk soil. More specifically, several potential PGPBs from Proteobacteria such as *Bradyrhizobium*, *Rhizobium* group, *Sphingomonas*, and Rhodanobacteraceae, and Actinobacteria *Streptomyces*, Micromonosporaceae, *Actinoplanes*, and *Nonomuraea* were relatively abundant in apple roots compared to soil. *Rhizobium* and *Bradyrhizobium* are well-known nitrogen fixers [65] and are relatively abundant in apple roots compared to orchard soil. Previously, it was reported that some *Sphingomonas* taxa could improve plant growth under stress by producing plant growth hormones such as gibberellins and IAA [66], while members of *Sphingomonas* taxa can produce antioxidant enzymes to improve host-plant stress adaptation. Members of Rhodanobacteraceae also can have an antagonistic effect against plant pathogens such as *Fusarium solani* and *Ralstonia solanacearum* [67]. Actinobacteria *Streptomyces* is a known taxon with the potential to improve host plant growth and produce ACC deaminase, IAA, and lytic enzymes [68], which are relatively more abundant in apple roots than in soil. 

Fungal orders Pleosporales and Helotiales were highly abundant in apple roots and were represented by >70% of the total ITS reads. These orders are comprised of both important plant pathogens and dark septate endophytes (DSEs). It is well documented that the plant-DSE association can improve plant host nutrient uptake and reduce stress tolerance [69]. On the other hand, the relative abundances of *Fusarium* and Nectriaceae were significantly decreased in apple plant roots compared to the orchard soil. *Fusarium* spp. are well-known plant pathogens, and species from the family Nectriaceae were recently identified as contributors to ARD [70]. These data support our hypothesis that the roots of mature apple trees might be colonized by a number of plant growth-promoting microorganisms that can reduce pathogen attacks on the plants. 

## 5. Conclusions

Currently, little is known about the post-plant development of the apple root microbiome as it relates to ARD. It is probable that the root-associated microbiome of the newly planted young tree is quickly affected by the existing microbiome in the old orchard soil, which is then subjected to change by strong sorting pressure and partially defined by environmental and biological factors/stresses faced by the tree. Ideally, this pressure shapes the root endophytic community as well, in a way that allows the microbiome to protect the plant from biotic and abiotic stresses. Our hypothesis is that young trees on clonal rootstocks of a specific genotype, when propagated conventionally in the nursery beds often treated with herbicides and pesticides to control weeds and pests or by micropropagation in tubes in a sterile environment when moved from nursery environments to an orchard replant site, have not had sufficient time or the opportunity to acquire a symbiosis with beneficial root endophytes present in their new environment. If this is true, the introduction of a synthetic microbial community composed of culturable endophytes isolated from mature, healthy trees in the plant propagation phase might be a way to “immunize” young trees and make them more resistant/tolerant to ARD. In this study, we identified a group of root endophytes that might be a target for isolation and testing for their potential to improve young tree adaptation to soils affected by ARD. However, a comprehensive evaluation of root microbiomes associated with nursery trees, as well as a deeper taxonomic resolution of mature and young tree root microbiomes, are necessary for a more specific composition for this “immunization” community.

## Figures and Tables

**Figure 1 microorganisms-11-01372-f001:**
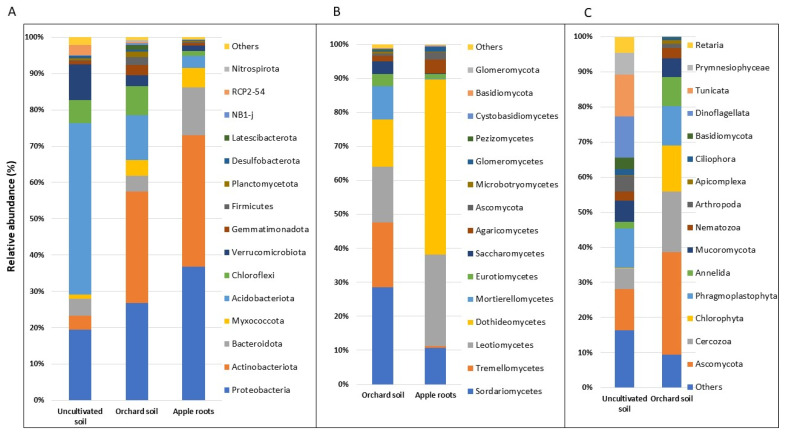
Relative abundances of (**A**) bacterial, (**B**) fungal, and (**C**) eukaryotic taxa were detected in apple root, cultivated orchard soil, and uncultivated bulk soil. Uncultivated bulk soil sample data were from Yurgel et al. [22]. Data of cultivated orchard soil and apple tree roots are referred to six mature apple orchards, each with 6 sub-samples.

**Figure 2 microorganisms-11-01372-f002:**
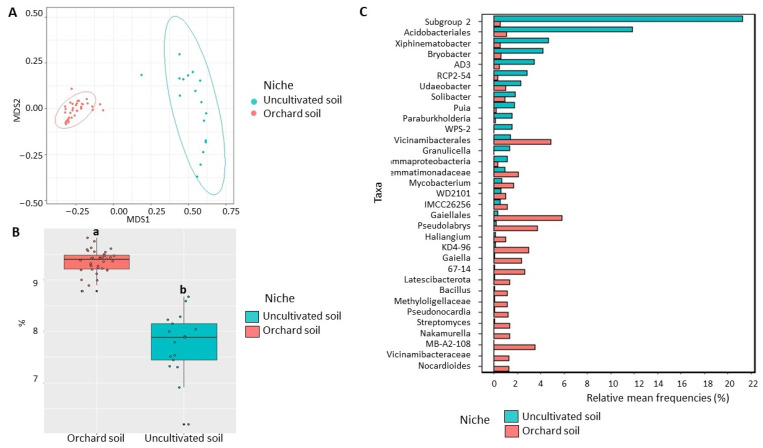
Diversity of bacterial communities in the orchard and uncultivated soil. (**A**) Non-metric multidimensional scaling (NMDS) based on Bray–Curtis distances. (**B**) Shannon diversity index (data followed by different letters are significantly different according to Kruskal–Wallis at *p* < 0.05. (**C**) Differentially represented bacterial taxa with a relative frequency > 1%. Data of uncultivated bulk soil samples were from Yurgel et al. [22]. Data of cultivated orchard soil were referred to six mature apple orchards, each with 6 sub-samples.

**Figure 3 microorganisms-11-01372-f003:**
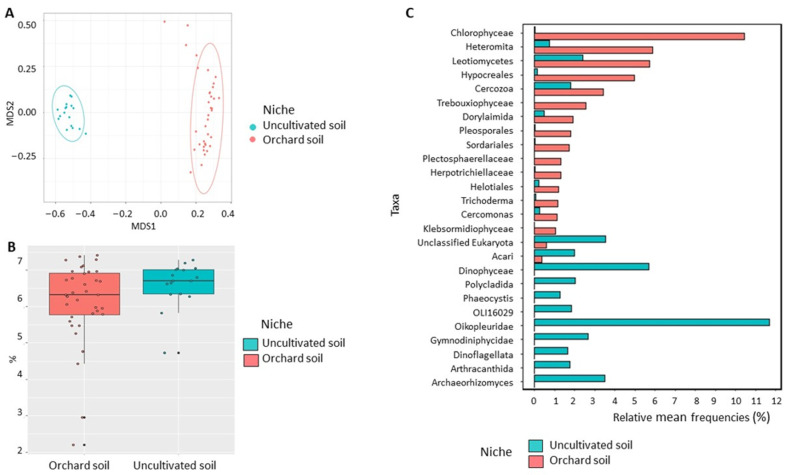
Diversity of eukaryotic communities in the orchard and uncultivated soil. (**A**) Non-metric multidimensional scaling (NMDS) based on Morisita-Horn dissimilarity indices. (**B**) Shannon diversity index according to Kruskal–Wallis test at *p* < 0.05. (**C**) Differentially represented eukaryotic taxa with a relative frequency >1%. Data of uncultivated bulk soil samples were from Yurgel et al. [22]. Data of cultivated orchard soil were referred to six mature apple orchards each with 6 sub-samples.

**Figure 4 microorganisms-11-01372-f004:**
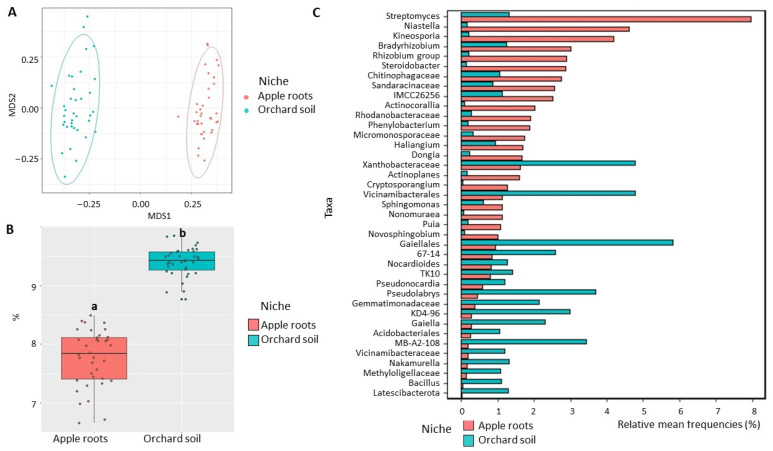
Diversity of bacterial communities in the apple roots and orchard soil. (**A**) Non-metric multidimensional scaling (NMDS) based on Bray–Curtis distances. (**B**) Shannon diversity index (data followed by different letters are significantly different according to Kruskal–Wallis at *p* < 0.05. (**C**) Differentially represented bacterial taxa with a relative frequency >1%. Data of cultivated orchard soil and apple tree roots were referred to six mature apple orchards, each with 6 sub-samples.

**Figure 5 microorganisms-11-01372-f005:**
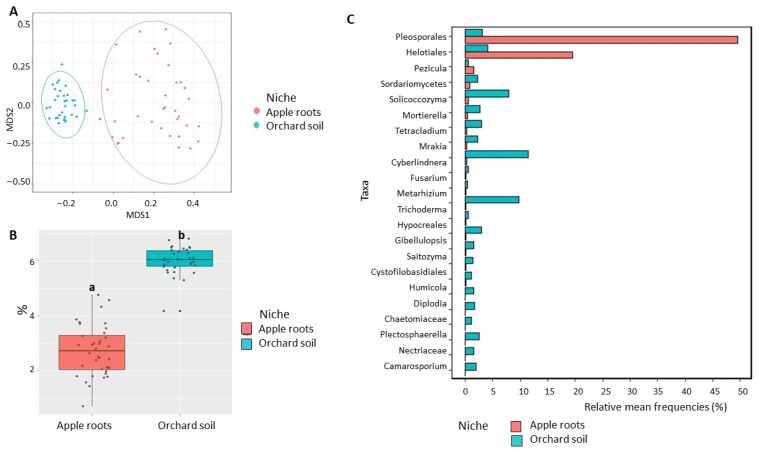
Diversity of fungal communities in the apple roots and orchard soil. (**A**) Non-metric multidimensional scaling (NMDS) based on Bray–Curtis distances. (**B**) Shannon diversity index (data followed by different letters are significantly different according to Kruskal–Wallis at *p* < 0.05. (**C**) Differentially represented fungal taxa with a relative frequency > 1%. Data of cultivated orchard soil and apple tree roots were referred to six mature apple orchards, each with 6 sub-samples.

**Figure 6 microorganisms-11-01372-f006:**
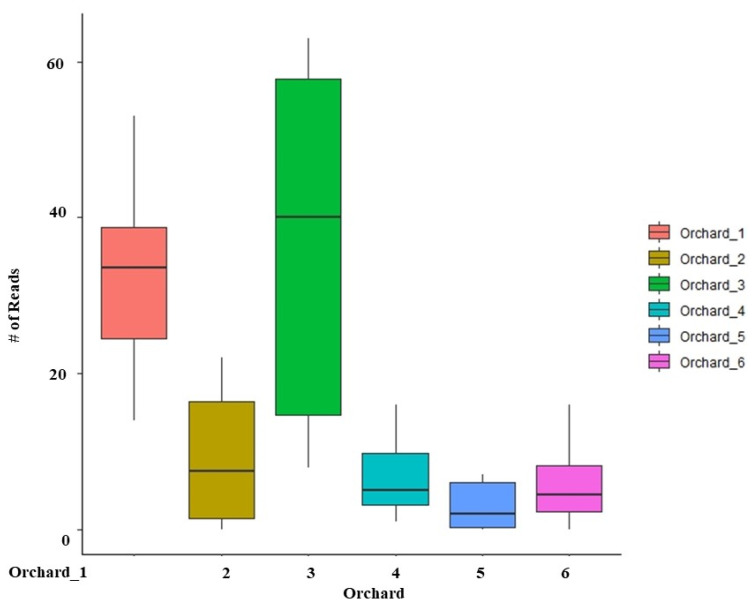
Relative abundance of *Trichoderma* across the orchards, Corrected *p*-values (q-values) were calculated based on Benjamini–Hochberg FDR multiple test correction. Features with Welch’s test *q* < 0.05 were considered significant.

**Table 1 microorganisms-11-01372-t001:** Variation in sample groupings as explained by Bray–Curtis distance matrix.

Type of Amplicon	Parameter	R^2^
16S rRNA		
Apple root/Orchard soil	Niche	0.305 ***
	Location	0.157 ***
	Niche × Location	0.104 ***
Orchard/Uncultivated soil	Niche	0.288 ***
Apple root	Location	0.358 ***
Orchard soil	Location	0.399 ***
ITS		
Apple root/Orchard soil	Niche	0.389 ***
	Location	0.142 ***
	Niche × Location	0.082 **
Apple root	Location	0.309 ***
Orchard soil	Location	0.424 ***
18S rRNA		
Orchard/Uncultivated soil	Niche	0.304 ***
Orchard soil	Location	0.341 ***

Adonis tests were used to assess whether beta-diversity is related to sample groupings, 999 permutations, R^2^, ** *p* < 0.01, *** *p* < 0.001.

## Data Availability

The datasets generated in the current study are available in the [SRA NCBI] repository and can be accessed from the following links https://www.ncbi.nlm.nih.gov/sra/PRJNA932770, https://www.ncbi.nlm.nih.gov/sra/PRJNA933105 and https://www.ncbi.nlm.nih.gov/sra/PRJNA933100 (accessed on 20 February 2023).

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
