# Peer review of "Apple Root Microbiome as Indicator of Plant Adaptation to Apple Replant Diseased Soils"

_microorganisms, 2023, doi:10.3390/microorganisms11061372_

Round 1
Reviewer 1 Report
General Comment:
The manuscript by Ajeethan et al. is a well written document which reports on studies that sought to characterize composition of the orchard soil and apple root microbiome. Furthermore, attempt was made to evaluate differences in these attributes between the orchard system and that detected in soil that had not previously been planted to apple. Aspects of these studies have been conducted previously in orchard systems at other locales in North America and Europe and should be noted in the text. Certain of the experimental details will require further detail and I have addressed these in my specific comments. There are also elements regarding the published literature and previous reports that have been overlooked, which directly relate to the studies described here. There should be some attempt to note these and perhaps incorporate into your discussion. I believe that the conclusions statement needs to be reassessed based on our knowledge of the apple root endophytic microbiome as constructed even in very young rootstock material.
Specific Comments:
Lines 72-78: I believe there should be additional information provided at this point regarding the effect of the host on their associated microbiome. The host plant genotype may shape their associated microbiome through differential production and release into the rhizosphere of specific root exudates (metabolites). Leisso et al. (2017; 2018) demonstrated that apple rootstock genotype differ in metabolic composition of root exudates; for example rootstock G.935 produces greater quantities of myo-inositol than M.26. Subsequently, it was demonstrated the myo-inositol exhibits a degree of repulsion towards Pratylenchus penetrans (Somera and Mazzola, 2022 Front. Microbiol.) and correspondingly G.935 was less susceptible to colonization by this lesion nematode than is M.26. Thus, the not only is the plant shaping its microbiome, it is deirected in a plant-genotype dependent manner.
Section 2.1: A further description of the sampling sites is required. What were the attributes of the undisturbed forest sites; forest cover type? Soil characteristics such as soil pH, soil type, etc.? These few soil characteristics in and of themselves will have significant effect on shaping the composition of the soil microbiome. How do these attributes differ from the orchard sites? Were these apple trees really the age reported in the text (100 years old?) or was this how long the orchards had been in place? What were the tree cultivars and what rootstock employed? As rootstock genotype/cultivar will have a significant impact on the rhizosphere/endophytic microbiome (Van Horn et al. 2021 Phytobiomes) this is an important element to report and consider in drafting your conclusions.
Line 130-131. Is it correct to conclude that the sequence data from the ‘uncultivated soil’ were the data generated and reported in the 2019 manuscript (19)? And thus, the sequencing data obtained for the forest soils and that obtained from the orchard soils were not derived from the same sequencing runs. If so, this is problematic as minor differences in the sequencing run, whether the platform used or the analysis pipeline or any physical alteration in processing anywhere along the line, can have significant impact on the assignment of amplicons to a particular ASV. I am not at all certain as to whether this can be remedied at this point and could impact the findings and assumptions as detailed in this report. As such, one would need to question the validity of making any comparison between the forest and orchard soil microbiomes as reported here.
Line 346; Line 355: The statement that host plant and genotype can direct composition of the root-associated microbiome and yet there is no reference to the work in apple that has previously reported this statement. Rather the work cited references the beech tree and a wild perennial. The effect of apple rootstock genotype on the root and rhizosphere microbiome has been documented repeatedly (see Liu et al. Microbiome 2018; Van Horn et al. Phytobiomes 2021; Wang and Mazzola, 2019, Phytopathology 109:607-614; Deakin et al. Phytobiomes 2019). These important works on the apple microbiome should be viewed and cited.
Line 390 and elsewhere in the text: Unfortunately, recent work from China has filled the literature with unsubstantiated reports of Fusarium spp. as causal agents of replant disease. This, regardless of the fact that a number of studies demonstrated that numerous Fusarium spp. are not pathogenic toward apple. The work that you cite regarding Fusarium proliferatum as a replant pathogen based the assessment of this fungus as such on a singular study reported in Plant Disease (Duan et al. https://doi.org/10.1094/PDIS-12-21-2802-RE). In this 2022 manuscript, it can be seen in the photos that the F. proliferatum was not inducing replant symptoms at all even though symptoms of wilt or dessication were apparent. In addition, the fact that Aspergillus flavus and Alternaria alternata were inducing precisely the same symptoms (and have never been reported as root pathogens of any plant) should have brought into question the validity of the experimental methods used and the report conclusions that F. proliferatum was a replant pathogen. Consider these statements when citing Fusarium as a pathogen inciting replant disease.
Line 419-421: The findings of significant variation in ‘structure, composition and diversity” between the apple root and soil microbiome were previously reported in the work of Van Horn et al. 2021. The comparison of the root endophytic microbiome and rhizosphere soil microbiome were reported with virtually the same findings reported here in terms of these attributes. This should be noted.
Discussion: The authors base many of there conclusions on microbial IDs reported on a level that make it difficult to report clear function. This has been a repeated limitation to assessing actionable items based upon evaluation of amplicon sequencing data that defines compositional attributes. Merely stating that “A” is different from “B” (orchard soil is different from forest soil in terms of microbiome composition) is repeatedly being reported but does not relay to us the importance of the difference and how to proceed in terms of management. As such, the need to employ other methodologies to ascertain functional attributes of the microbiome has been recognized. As an example, Pseudomonas or Fusarium spp. can function as pathogens, endophytes or as biocontrol agents. Making statements of function based upon phyla or Genus level assignments of ID should be limited in the discussion.
Conclusion: I believe the authors should reassess their statement of conclusion, specifically lines 458-464. The hypothesis as stated is not valid. Apple rootstocks and trees produced in a commercial nursery typically have inhabited that soil environment for multiple years; commonly 2-3 years or longer. The vast majority have already established a robust endophytic community prior to out-planting these trees into an orchard system. Results from examining the apple root endophytic microbiome in 6-12 week pot trials, an obviously shorter duration than a nursery grown plant, demonstrate this fact where trees rapidly establish a robust endophytic mycorrhizal populations; when apple rootstocks were planted into two different orchard soils, mycorrhizal fungi represented 14 to 17% of the total fungal OTUs detected in the apple root endophytic environment (Van Horn et al. 2021). If the approach suggested by the authors to establish a ‘synthetic microbiome’ on this plant material were pursued, such a method would need to be employed far earlier in the plant propagation phase…not at orchard establishment…in order to realize any potential success.
Author Response
Lines 72-78: I believe there should be additional information provided at this point regarding the effect of the host on their associated microbiome. The host plant genotype may shape their associated microbiome through differential production and release into the rhizosphere of specific root exudates (metabolites). Leisso et al. (2017; 2018) demonstrated that apple rootstock genotype differ in metabolic composition of root exudates; for example rootstock G.935 produces greater quantities of myo-inositol than M.26. Subsequently, it was demonstrated the myo-inositol exhibits a degree of repulsion towards Pratylenchus penetrans (Somera and Mazzola, 2022 Front. Microbiol.) and correspondingly G.935 was less susceptible to colonization by this lesion nematode than is M.26. Thus, the not only is the plant shaping its microbiome, it is deirected in a plant-genotype dependent manner.
Thank you for your suggestion to add additional information. Influence of plant metabolite/ plant genotype on microbiome is added in the introduction part with new references.
Section 2.1: A further description of the sampling sites is required. What were the attributes of the undisturbed forest sites; forest cover type? Soil characteristics such as soil pH, soil type, etc.? These few soil characteristics in and of themselves will have significant effect on shaping the composition of the soil microbiome. How do these attributes differ from the orchard sites? Were these apple trees really the age reported in the text (100 years old?) or was this how long the orchards had been in place? What were the tree cultivars and what rootstock employed? As rootstock genotype/cultivar will have a significant impact on the rhizosphere/endophytic microbiome (Van Horn et al. 2021 Phytobiomes) this is an important element to report and consider in drafting your conclusions.
That is a good question. Soil pH was added in section 2.1. The Uncultivated soil was sandy loam. But Orchard soil has apple trees only. Uncultivated soil samples were collected from lands with more diverse plants. We used that previous study information here to compare the orchard soil. Their physio- chemical properties were published earlier, but we did not have orchard soil properties to compare.
Instead of “age” sentence is replaced as “Sampling location and orchard age." Sorry for the confusion.
Line 130-131. Is it correct to conclude that the sequence data from the ‘uncultivated soil’ were the data generated and reported in the 2019 manuscript (19)? And thus, the sequencing data obtained for the forest soils and that obtained from the orchard soils were not derived from the same sequencing runs. If so, this is problematic as minor differences in the sequencing run, whether the platform used or the analysis pipeline or any physical alteration in processing anywhere along the line, can have significant impact on the assignment of amplicons to a particular ASV. I am not at all certain as to whether this can be remedied at this point and could impact the findings and assumptions as detailed in this report. As such, one would need to question the validity of making any comparison between the forest and orchard soil microbiomes as reported here.
It is added in the discussion section “Additionally, uncultivated soil samples were collected, processed, and sequenced previous year. There might be some minor differences which could affect slightly the ASV annotation. However, this factor would not introduce changes in forest soil microbiome comparable to the differences resulted by factors like soil origin or management. While considering the differences between orchard and forest soil it is also important to emphasize that the soil properties might also be a factor introducing the differences between these two types of soil. For example, soil pH might influence microbiome [40]. Nevertheless, in this study we are focusing on the major characteristics of orchard soils, such as presence of plant-growth promoting and pathogenic microorganisms."
Additionally, there are a number of large scale data analysis studies incorporating sequences obtained by separate runs, which proven to be reliable. Please see an example: Nearing et. al., 2022, Nat commun. Microbiome differential abundance methods produce different results across 38 datasets
Line 346; Line 355: The statement that host plant and genotype can direct composition of the root-associated microbiome and yet there is no reference to the work in apple that has previously reported this statement. Rather the work cited references the beech tree and a wild perennial. The effect of apple rootstock genotype on the root and rhizosphere microbiome has been documented repeatedly (see Liu et al. Microbiome 2018; Van Horn et al. Phytobiomes 2021; Wang and Mazzola, 2019, Phytopathology 109:607-614; Deakin et al. Phytobiomes 2019). These important works on the apple microbiome should be viewed and cited.
Thank you for your suggestion to add these references. Two of those references were removed and instead of them some new references are added in the discussion part according to your suggestion.
Line 390 and elsewhere in the text: Unfortunately, recent work from China has filled the literature with unsubstantiated reports of Fusarium spp. as causal agents of replant disease. This, regardless of the fact that a number of studies demonstrated that numerous Fusarium spp. are not pathogenic toward apple. The work that you cite regarding Fusarium proliferatum as a replant pathogen based the assessment of this fungus as such on a singular study reported in Plant Disease (Duan et al. https://doi.org/10.1094/PDIS-12-21-2802-RE). In this 2022 manuscript, it can be seen in the photos that the F. proliferatum was not inducing replant symptoms at all even though symptoms of wilt or dessication were apparent. In addition, the fact that Aspergillus flavus and Alternaria alternata were inducing precisely the same symptoms (and have never been reported as root pathogens of any plant) should have brought into question the validity of the experimental methods used and the report conclusions that F. proliferatum was a replant pathogen. Consider these statements when citing Fusarium as a pathogen inciting replant disease.
The texts related to Fusarium is reconstructed according to the suggestion to increase the validity of the text.
Line 419-421: The findings of significant variation in ‘structure, composition and diversity” between the apple root and soil microbiome were previously reported in the work of Van Horn et al. 2021. The comparison of the root endophytic microbiome and rhizosphere soil microbiome were reported with virtually the same findings reported here in terms of these attributes. This should be noted.
Thanks for your suggestion, reference is added.
Discussion: The authors base many of there conclusions on microbial IDs reported on a level that make it difficult to report clear function. This has been a repeated limitation to assessing actionable items based upon evaluation of amplicon sequencing data that defines compositional attributes. Merely stating that “A” is different from “B” (orchard soil is different from forest soil in terms of microbiome composition) is repeatedly being reported but does not relay to us the importance of the difference and how to proceed in terms of management. As such, the need to employ other methodologies to ascertain functional attributes of the microbiome has been recognized. As an example, Pseudomonas or Fusarium spp. can function as pathogens, endophytes or as biocontrol agents. Making statements of function based upon phyla or Genus level assignments of ID should be limited in the discussion.
We are presenting our results in figures (2,3,4,5) at the level-6 (species), due to this reason we are discussing at level-6. We will go deeper to level -7 in our upcoming publications. Thank you for your great suggestion.
Conclusion: I believe the authors should reassess their statement of conclusion, specifically lines 458-464. The hypothesis as stated is not valid. Apple rootstocks and trees produced in a commercial nursery typically have inhabited that soil environment for multiple years; commonly 2-3 years or longer. The vast majority have already established a robust endophytic community prior to out-planting these trees into an orchard system. Results from examining the apple root endophytic microbiome in 6-12 week pot trials, an obviously shorter duration than a nursery grown plant, demonstrate this fact where trees rapidly establish a robust endophytic mycorrhizal populations; when apple rootstocks were planted into two different orchard soils, mycorrhizal fungi represented 14 to 17% of the total fungal OTUs detected in the apple root endophytic environment (Van Horn et al. 2021). If the approach suggested by the authors to establish a ‘synthetic microbiome’ on this plant material were pursued, such a method would need to be employed far earlier in the plant propagation phase…not at orchard establishment…in order to realize any potential success.
Thank you for the suggestion. We made a reference to in the plant propagation phase as an important time point for “immunization”
Reviewer 2 Report
The introduction introduces the issue well, but in my opinion you need to reformulate the hypothesis, because here we do not have plant roots differing in health, only the selection of endophytes from the soil environment is checked. The hypotheses should be about the variants being compared, as the results are arranged. The results should also compare the communities more rather than describing individually what was identified on each variant, which would shorten the description considerably. The discussion doesn't compare the differences between the variants too much, but instead focuses on explaining the role individual taxa can play in the occupied niche. The discussion should be based on the hypotheses raised.
Author Response
The introduction introduces the issue well, but in my opinion you need to reformulate the hypothesis, because here we do not have plant roots differing in health, only the selection of endophytes from the soil environment is checked. The hypotheses should be about the variants being compared, as the results are arranged. The results should also compare the communities more rather than describing individually what was identified on each variant, which would shorten the description considerably. The discussion doesn't compare the differences between the variants too much, but instead focuses on explaining the role individual taxa can play in the occupied niche. The discussion should be based on the hypotheses raised.
Thank you very much for your suggestion, some sentences are reconstructed. The hypothesis and discussion are also modified. Some new references are added, and the figures are replaced with more clear figures.
Reviewer 3 Report
If I correctly read the paper microorganisms-2308691-peer-review-v1 “Apple root microbiome as indicator of plant adaptation to Apple Replant Diseased soils”, the authors report interesting data on Apple replant disease (ARD) in Nova Scotia. 16S rRNA/18S rRNA and 16S 15 rRNA/ITS2 amplicon sequencing were used to identify bacterial and fungal communities associated with soil and root microbiomes. Uncultivated bulk soil samples were collected in August 2015, while orchard bulk soil (cultivated) and apple root samples were collected between September 2019 and September 2020 from six mature apple orchards. Bioinformatics and statistical analysis indicated significant (p<0.05) differences in structure and composition of soil microbial communities between uncultivated and apple orchard soils. Apple orchard soil showed an increase in the number of pathogens associate with ARD development. On the other hand, orchard soil abounded in several potential plant-growth promoting (PGP) microorganisms, biocontrol agents and non-fungal eukaryotes useful to promoting the proliferation of bacterial biocontrol agents. Compared to the soil microbiome, apple root microbiome accumulated potential PGP bacteria, and decreased the relative abundances of Fusarium and Nectriaceae, the fungal taxa associated with ARD.
Notwithstanding the scientific sound of this work, the presentation in the form of a manuscript is confusing, unclear, and requires adjustments. Information on the replant disease is mixed with the Apple replant disease.
The title is not fully pertinent to the topics of the manuscript.
The abstract and the conclusion are two very important parts of the manuscript; I suggest rewriting them.
Does it make sense to compare the microbiome of 2015 with that collected in 2019-2020?
Use pertinent keywords.
Insert the Authors name to all the cited organisms or delete “[Suckow] 11 Borkh” on line 11.
In non-taxonomic papers, it is facultative to indicate the Authors associated with Latin names of plants, fungi, and organisms.
Lines 37-38: the period is not clear. Rewrite.
Lines 41-44: the period is not clear. Rewrite.
Line 50: What does “ARD” mean?
Lines 55-59: the period is not clear. Rewrite.
Line 74: What does “PGP” mean?
Lines 79-90: the periods are not clear. Rewrite.
Lines 103-100: the periods are not clear. Rewrite.
Line 108: is it necessary to indicate the full name of DNA?
Lines 121-123: usually primers have the indication 5' -3'
The whole results session is presented in a confusing manner.
Figure 1: improve the magnification. If possible, use different colours for the taxa in the three sections of the figure. The current choice of colours creates a lot of confusion and makes the figure illegible.
Lines 179-182: the figure 1 legend is not clear. Rewrite.
Perhaps three figures would help the reader in following the differences of the three analysed communities.
Figure 2: improve the magnification of the C section.
Lines 235-242: the figure legend is not clear. Rewrite.
Figure 3: improve the magnification of the C section.
Lines 268-275: the figure legend is not clear. Rewrite.
Figure 4: improve the magnification of the C section.
Lines 296-302: the figure legend is not clear. Rewrite.
Figure 5: improve the magnification of the C section.
Lines 318-325: the figure legend is not clear. Rewrite.
Lines 340-342: the figure legend is not clear. Rewrite.
Improve the discussion section.
Line 344: What does “sessile” mean?
Kind regards
Author Response
Notwithstanding the scientific sound of this work, the presentation in the form of a manuscript is confusing, unclear, and requires adjustments. Information on the replant disease is mixed with the Apple replant disease.
Sorry for the confusion. We made some modifications to the paper to clarify the issue.
The title is not fully pertinent to the topics of the manuscript.
The abstract and the conclusion are two very important parts of the manuscript; I suggest rewriting them.
Thanks. Both abstract and conclusion are rewritten.
Does it make sense to compare the microbiome of 2015 with that collected in 2019-2020?
It is added in the discussion section “Additionally, uncultivated soil samples were collected, processed, and sequenced previous year. There might be some minor differences which could affect slightly the ASV annotation. However, this factor would not introduce changes in forest soil microbiome comparable to the differences resulted by factors like soil origin or management. While considering the differences between orchard and forest soil it is also important to emphasize that the soil properties might also be a factor introducing the differences between these two types of soil. For example, soil pH might influence microbiome [40]. Nevertheless, in this study we are focusing on the major characteristics of orchard soils, such as presence of plant-growth promoting and pathogenic microorganisms."
Use pertinent keywords.
Added.
Insert the Authors name to all the cited organisms or delete “[Suckow] 11 Borkh” on line 11.
In non-taxonomic papers, it is facultative to indicate the Authors associated with Latin names of plants, fungi, and organisms.
Removed
Lines 37-38: the period is not clear. Rewrite.
Restructured
Lines 41-44: the period is not clear. Rewrite.
Restructured
Line 50: What does “ARD” mean?
Added “Apple Replant Disease”
Lines 55-59: the period is not clear. Rewrite.
Restructured
Line 74: What does “PGP” mean?
Added “Plant Growth Promoting “
Lines 79-90: the periods are not clear. Rewrite.
Lines 103-100: the periods are not clear. Rewrite.
Restructured
Line 108: is it necessary to indicate the full name of DNA?
Removed
Lines 121-123: usually primers have the indication 5' -3'
Added
The whole results session is presented in a confusing manner.
Sorry for the confusion, first we are describing about our data (3.1) for bacteria, fungi and eukaryotes, then we are comparing between soil microbiome (3.2) (bacteria and eukaryote) and then we are comparing apple root and soil microbiome (3.3) (bacteria and fungi). Finally, we are describing about soil and root microbiome changes along the orchards (3.4).
Figure 1: improve the magnification. If possible, use different colours for the taxa in the three sections of the figure. The current choice of colours creates a lot of confusion and makes the figure illegible. Perhaps three figures would help the reader in following the differences of the three analysed communities.
Sorry for the confusion, each panel was separated by the boarders to easy differentiation. We are presenting data with same analysis for all 3 niches in a single look for bacteria, fungal and eukaryotes communities. That’s why we presented in a single figure.
Lines 179-182: the figure 1 legend is not clear. Rewrite.
Figure 2: improve the magnification of the C section.
Done.
Lines 235-242: the figure legend is not clear. Rewrite.
Figure 3: improve the magnification of the C section.
Done.
Lines 268-275: the figure legend is not clear. Rewrite.
Figure 4: improve the magnification of the C section.
Done.
Lines 296-302: the figure legend is not clear. Rewrite.
Figure 5: improve the magnification of the C section.
Done.
Lines 318-325: the figure legend is not clear. Rewrite.
Lines 340-342: the figure legend is not clear. Rewrite.
Improve the discussion section.
Some sentences were reconstructed, and some references were added.
Line 344: What does “sessile” mean?
We try to say plants that do not move in their environment.
Round 2
Reviewer 3 Report
The revised version was improved but remains confusing, unclear, and repetitive.
The title remains not fully pertinent to the topics of the manuscript.
Improve the abstract.
The introduction is in some parts repetitive. See lines 58-60 and 78-80.
The Material and Methods section remains confusing.
If soil pH is important, insert the procedure in Material and Methods and the recorded value for all collected orchard soil sub-samples.
Line 125: What does “washed several times” mean?
Line 125: What does “sonicated several times” mean?
Line 127: What does “root tissue” mean? After liquid nitrogen treatment, it is impossible to recognize root tissue.
Line 127: What does “isolation” mean? Use “extraction”
Lines 156-157: use “amplicon sequence variants (ASVs) high-resolution genomic groupings” instead of “amplicon sequence variants (ASVs-high resolution genomic groupings)”
Section 2.4. How were data normalized?
Line 175: What does “produced” mean?
If I correctly read the paper, the following arrangements for the Material and Methods section are suggested:
2.1. Site and sample collection
(Avoid lines 105-109)
2.2. DNA extraction and sequencing
2.3. Sequencing data processing and statistical analysis
(Insert information from lines 105-109)
The Results section remains confusing.
Avoid information on Material and Methods
Verify the correct name of the cited phyla.
Until now, Fungi are Eukaryotic organisms. Why were they separated?
Uniform the data analyses. Section 3.1 considers Bacterial, Fungal and Eukaryotic communities, while section 3.2 describes Bacterial and Eukaryotic communities.
A possible suggestion for data presentation:
3.1. Taxonomic profiles of soil and root associated microbiomes
3.1.1. Bacterial communities
3.1.2. Eukaryotic communities (including fungi)
3.1.2.1. Fungal communities (if not considered in the Eukaryotic communities)
3.2. Comparisons of microbial communities
3.2.1. Uncultivated vs orchard soil
3.2.1.1 Bacterial communities
3.2.1.2. Eukaryotic communities (including fungi)
3.2.1.3. Fungal communities (if not considered in the Eukaryotic communities)
3.2.2. Apple root vs orchard soil
3.2.2.1 Bacterial communities
3.2.2.2. Eukaryotic communities (including fungi)
3.2.2.3. Fungal communities (if not considered in the Eukaryotic communities)
Figures must be self-explanatory, clear, and easy to understand without needing any extra explanation.
Figure 1:
If possible, use a unique colour scale for all three sections.
Present the data in the sequence: A) Bacteria, B) Eukaryotes, C) Fungi.
Each section shows histograms for Uncultivated soil, Orchard soil and Apple roots.
Lines 198-200: The Figure legend is not clear. Rewrite. A possible suggestion:
Figure 1. Relative abundances of bacterial (A) and eukaryotic (B) phyla and fungal classes (C) detected in apple root, cultivated orchard soil, and uncultivated bulk soil. Uncultivated bulk soil sample data were from Yurgel et al. [22]. Data of cultivated orchard soil and apple tree roots are referred to six mature apple orchards each with 6 sub-samples.
Lines 210 and 219: What does “Figure S1” mean?
Use “Figure S” or “Supplementary Figure S”. Uniform.
Figure 2 C:
Improve magnification.
Taxa including genera (Bacillus, Pseudonocardia, Streptomyces, Nakamurella, Nocardioides), families (67-41, Methyloligellaceae, Vicinamibacteraceae), classes (MB-A2-108) and Phylum (Latescibacterota), while the legend considers “genera”.
Use “Relative mean frequencies (%)” instead of “Relative mean frequencies %”
If it is possible, add indications on “other taxa” frequencies.
Lines 260-265: the legend is not clear. Rewrite. A possible suggestion:
Figure 2. Diversity of bacterial communities in the orchard and uncultivated soil. A) Non-metric multidimensional scaling (NMDS) based on Bray-Curtis distances. B) Shannon diversity index (data followed by different letters are significantly different according to Kruskal-Wallis at p < 0.05. C) Differentially represented bacterial taxa with a relative frequency >1%. Data of uncultivated bulk soil samples were from Yurgel et al. [22]. Data of cultivated orchard soil and apple tree roots are referred to six mature apple orchards each with 6 sub-samples.
Table 1:
Improve table descriptive title.
Insert opportune indications for table footers.
Lines 241 and 540: Where is “Table S2”?
Lines 277 and 541: Where is “Table S3”?
Lines 289 and 542: Where is “Table S4”?
Lines 307 and 543: Where is “Table S5”?
Line 331: Where is “Table S6”?
Figures 3, 4, 5 and 6:
Legends are not pertinent. Rewrite.
Improve the discussion section.
Improve the Conclusion section.
It was very difficult to read the pdf file in revision form. If possible, highlight the modified parts in yellow.
Kind regards
Author Response
Reviewer_3:
The title remains not fully pertinent to the topics of the manuscript. We believe that the title is adequate unless the reviewer has a better suggestion.
Improve the abstract. Done
The introduction is in some parts repetitive. See lines 58-60 and 78-80. We did not see any repetition in the sentences below: lines 58-60: Multiple strategies have been applied to comprehensively understand ARD etiology. Culturing of apple trees can trigger changes in indigenous soil microbiome that may lead to disease development. And lines 78-80: Management practices to control ARD can be costly, inefficient and in many cases, detrimental to human and environmental health, and soil sustainability.
The Material and Methods section remains confusing. We did our best to make it more clear.
If soil pH is important, insert the procedure in Material and Methods and the recorded value for all collected orchard soil sub-samples.We have done the physiochemical property analysis for uncultivated soil only Yurgel et al. [22], However, the orchard soil chemical properties were not analyzed. We will keep that in mind for future experiments. Thanks for this very important suggestion.
Line 125: What does “washed several times” mean? Corrected with thanks, here we make sure that other particles outside the root surface were removed and microbes associated with roots only undergone the extraction.
Line 125: What does “sonicated several times” mean? Corrected with thanks, here we make sure that dirt and debris were removed from the root surface and also it will help in DNA extraction process.
Line 127: What does “root tissue” mean? After liquid nitrogen treatment, it is impossible to recognize root tissue. But we believe it is still root tissue.
Line 127: What does “isolation” mean? Use “extraction” Thanks for your suggestion. Corrected.
Lines 156-157: use “amplicon sequence variants (ASVs) high-resolution genomic groupings” instead of “amplicon sequence variants (ASVs-high resolution genomic groupings)” Thanks for your suggestion, it is changed as you suggested.
Section 2.4. How were data normalized? Please see Results; Data description section, lines 171-173; 193-195 and 214-216.
Line 175: What does “produced” mean? Word “produced” is replaced in the paper as “The graphics were created through RStudio”.
If I correctly read the paper, the following arrangements for the Material and Methods section are suggested: We would like to keep the current format, which present the data better.
The Results section remains confusing. We did our best to make it more clear.
Avoid information on Material and Methods
Verify the correct name of the cited phyla. Done
Until now, Fungi are Eukaryotic organisms. Why were they separated? We used ITS2 amplicon sequencing for specially identify fungal communities. ITS is identified as a fungal genetic barcode because it is the most sequenced region of fungi and is routinely used for and identification (Begerow et al., 2010 https://link.springer.com/content/pdf/10.1007/s00253-010-2585-4.pdf and Bellemain et al., 2010 https://link.springer.com/article/10.1186/1471-2180-10-189). That’s the reason we used fungal communities analysis (ITS2 analysis) as separated other than eukaryotic communities (18s rRNA analysis).
Uniform the data analyses. Section 3.1 considers Bacterial, Fungal and Eukaryotic communities, while section 3.2 describes Bacterial and Eukaryotic communities.
A possible suggestion for data presentation: We would like to keep the current format, which present the data better.
Figure 1:
If possible, use a unique colour scale for all three sections. Done
Present the data in the sequence: A) Bacteria, B) Eukaryotes, C) Fungi. A) Bacteria, B) Fungi, C) Eukaryotes, it a better order, since it reflects data presentation
Each section shows histograms for Uncultivated soil, Orchard soil and Apple roots. There is not data for Apple root eukaryotes. It is difficult to sequence plant tissue with 18S rRNA to identify microorganisms because of interference with host 18S rRNA.
Lines 198-200: The Figure legend is not clear. Rewrite. A possible suggestion:
Figure 1. Relative abundances of bacterial (A) and eukaryotic (B) phyla and fungal classes (C) detected in apple root, cultivated orchard soil, and uncultivated bulk soil. Uncultivated bulk soil sample data were from Yurgel et al. [22]. Data of cultivated orchard soil and apple tree roots are referred to six mature apple orchards each with 6 sub-samples. Thank you very much for your suggestion. Legends for figure 1 was rewritten according to your suggestion.
Lines 210 and 219: What does “Figure S1” mean? Sorry for the confusion. Figure S1 is Supplementary Figure 1.
Use “Figure S” or “Supplementary Figure S”. Uniform. Thanks for the suggestion. Throughout the manuscript we are using Figure S, but for the first supplementary table and figure only denoted as “Supplementary Figure S1” and “Supplementary Table S1” other than that all the supplementary tables and figures are denoted as “Figure S and Table S”.
Figure 2 C: Improve magnification. Taxa including genera (Bacillus, Pseudonocardia, Streptomyces, Nakamurella, Nocardioides), families (67-41, Methyloligellaceae, Vicinamibacteraceae), classes (MB-A2-108) and Phylum (Latescibacterota), while the legend considers “genera”. Use “Relative mean frequencies (%)” instead of “Relative mean frequencies %” If it is possible, add indications on “other taxa” frequencies. We already improver magnification. We will work with the production team if more magnification is necessary. There is no “other taxa” in the fig 2. Thanks we changed “Relative mean frequencies (%)” as you suggested.
Lines 260-265: the legend is not clear. Rewrite. A possible suggestion: Figure 2. Diversity of bacterial communities in the orchard and uncultivated soil. A) Non-metric multidimensional scaling (NMDS) based on Bray-Curtis distances. B) Shannon diversity index (data followed by different letters are significantly different according to Kruskal-Wallis at p < 0.05. C) Differentially represented bacterial taxa with a relative frequency >1%. Data of uncultivated bulk soil samples were from Yurgel et al. [22]. Data of cultivated orchard soil and apple tree roots are referred to six mature apple orchards each with 6 sub-samples. Thank you very much for your suggestion. Legend was corrected according to your suggestion.
Table 1: Improve table descriptive title. Insert opportune indications for table footers. The believe the title is adequate.
Lines 241 and 540: Where is “Table S2”?
Sorry for the confusion. “Table S2. Bacterial class that showed significant differences in relative abundance in orchard and uncultivated soil” is a supplementary table, it is with supplementary files.
Lines 277 and 541: Where is “Table S3”? Sorry for the confusion. “Table S3. Eukaryotic class that showed significant differences in relative abundance in orchard and uncultivated soil” is a supplementary table, it is with supplementary files.
Lines 289 and 542: Where is “Table S4”? Sorry for the confusion. “Table S4. Causal agents of ARD and potential pathogenic fungi that showed differences in relative abundance in orchard soil and uncultivated soil” is a supplementary table, it is with supplementary files.
Lines 307 and 543: Where is “Table S5”? Sorry for the confusion. “Table S5. Bacterial class that showed significant differences in relative abundance in apple roots and orchard soil” is a supplementary table, it is with supplementary files.
Line 331: Where is “Table S6”? Sorry for the confusion. “Table S6. Fungal class that showed significant differences in relative abundance in apple roots and orchard soil” is a supplementary table, it is with supplementary files.
Figures 3, 4, 5 and 6: Legends are not pertinent. Rewrite. Thank you very much for your suggestion. Legends for figure 1,2,3,4 and 5 were rewritten according to your suggestion.
Improve the discussion section. We did our best to improve it.
Improve the Conclusion section. We did our best to to improve it.
It was very difficult to read the pdf file in revision form. If possible, highlight the modified parts in yellow.
Sorry for the inconvenience which we made for you; we are keeping your suggestion in mind and following it. Thanks.
